# Antimicrobial Activity and Mode of Action of Celastrol, a Nortriterpen Quinone Isolated from Natural Sources

**DOI:** 10.3390/foods10030591

**Published:** 2021-03-11

**Authors:** Nayely Padilla-Montaño, Leandro de León Guerra, Laila Moujir

**Affiliations:** Departamento de Bioquímica, Microbiología, Biología Celular y Genética, Facultad de Farmacia, Universidad de La Laguna, Avenida Astrofísico Fco Sánchez s/n, 382016 Tenerife, Spain; nayelypadilla@hotmail.com (N.P.-M.); lleongue@ull.edu.es (L.d.L.G.)

**Keywords:** celastrol, Celastraceae, antimicrobial activity, mechanism of action, *Bacillus subtilis*

## Abstract

Species of the Celastraceae family are traditionally consumed in different world regions for their stimulating properties. Celastrol, a triterpene methylene quinone isolated from plants of celastraceas, specifically activates satiety centers in the brain that play an important role in controlling body weight. In this work, the antimicrobial activity and mechanism of action of celastrol and a natural derivative, pristimerin, were investigated in *Bacillus subtilis*. Celastrol showed a higher antimicrobial activity compared with pristimerin, being active against Gram-positive bacteria with minimum inhibitory concentrations (MICs) that ranged between 0.16 and 2.5 µg/mL. Killing curves displayed a bactericidal effect that was dependent on the inoculum size. Monitoring of macromolecular synthesis in bacterial populations treated with these compounds revealed inhibition in the incorporation of all radiolabeled precursors, but not simultaneously. Celastrol at 3 µg/mL and pristimerin at 10 µg/mL affected DNA and RNA synthesis first, followed by protein synthesis, although the inhibitory action on the uptake of radiolabeled precursors was more dramatic with celastrol. This compound also caused cytoplasmic membrane disruption observed by potassium leakage and formation of mesosome-like structures. The inhibition of oxygen consumption of whole and disrupted cells after treatments with both quinones indicates damage in the cellular structure, suggesting the cytoplasmic membrane as a potential target. These findings indicate that celastrol could be considered as an interesting alternative to control outbreaks caused by spore-forming bacteria.

## 1. Introduction

Historically, plants have provided a wide variety of active compounds, becoming a key element in the development of many cultures. The Celastraceae family, commonly known as the bittersweet family, comprises a group of plants distributed mainly in tropical and subtropical regions of the world including North Africa, South America, and East Asia. It has a long history in traditional medicine as a stimulant, diuretic, emmenagogue, anti-inflammatory, antibacterial, anti-cancer, and for the treatment of gastrointestinal disorders among others [1]. In the Canary Islands, East Africa, and the Arabian Peninsula, the leaves of members of this family are chewed to combat fatigue [1,2]. In our search for antimicrobial compounds from plants, we isolated celastrol and its methylated derivative, pristimerin, which are the first and the most frequently reported celastroloids. The term celastroloid refers to methylene quinone nor-triterpenes with a 24-nor-*D: A*-friedo-oleanane skeleton, which, later on, was extended to related phenolic nor-triterpenes [3] and their dimer and trimer congeners. The two natural pentacyclic triterpenoids celastrol and pristimerin are commonly found in the roots and bark of Celastraceae species. Both compounds show a wide range of pharmacological activities, including anti-cancer, anti-inflammatory, antioxidant, hepatoprotective, or immunomodulatory, among others [4,5,6,7,8]. The anti-cancer properties of celastrol, one of the most studied methylene quinones, have been attributed to apoptosis and autophagy induction, cell cycle arrest, and anti-metastatic and anti-angiogenic action [9,10]. Other works have evidenced the ability of celastrol in combating metabolic disorders such as obesity and type 2 diabetes [11]. Treatment with celastrol suppresses food intake, blocks reduction in energy consumption, and mediates weight loss by acting as a leptin sensitizer in mouse models [12,13].

In addition to these pharmacological activities, antimicrobial properties have also been demonstrated for these and other related triterpenoids such as tingenone, netzahualcoyone, or zeylasterone that exhibit inhibitory activity against Gram-positive bacteria [14,15,16,17]. Moreover, the anti-biofilm [18,19,20] and anti-fouling properties of celastrol and pristimerin have been reported on a wide variety of microorganisms [21], as well as anti-mycotic activity [22,23,24,25,26,27]. Besides, celastrol and its derivatives have also shown inhibitory activity against viruses producing hepatitis B and C [28,29], which makes them potentially useful compounds for the control of various diseases. This paper describes the antimicrobial activity and mechanism of action of celastrol and pristimerin using *Bacillus subtilis* as a model of spore-forming bacteria. Species of *Bacillus* have been associated with many food contaminations causing food-borne illness in humans [30], spoilage of processed food products [31,32], or certain pathologies such as pneumonia, bacteremia, and meningitis in immunosuppressed patients [33,34,35]. Regardless of variations in disease presentation, the etiologic agent is often the spore and the production of toxins that play a central role in the pathophysiology of the infection. Although it is generally accepted that the antimicrobial activity of terpenoid compounds involves damage on plasma membrane [36], little is known about how this affects other cellular processes essential to bacterial cell development. Thus, the action of these compounds on different metabolic pathways such as macromolecular synthesis, uptake of solutes and biosynthetic precursors, or the damage on membrane function was investigated.

## 2. Materials and Methods

### 2.1. Microorganisms

Strains used for determining antimicrobial activity included *Bacillus subtilis* ATCC 6051, *B. cereus* ATCC 21772, *B. megaterium* ATCC 25848, *B. pumilus* ATCC 7061, *Staphylococcus aureus* ATCC 6538, *S. epidermidis* ATCC 14990, *S. saprophyticus* ATCC 15305, *Enterococcus faecalis* CECT 795 (from Type Culture Spanish Collection), *Mycobacterium smegmatis* ATCC 19420, *Proteus mirabilis* CECT170, *Escherichia coli* ATCC 9637, *Pseudomonas aeruginosa* AK958 (from the University of British Columbia, Department of Microbiology collection), *Salmonella* spp. CECT 456, and *Candida albicans* CECT 1039.

Bacterial cultures were developed at 37 °C in nutrient broth (NB), except for *E. faecalis* and *M. smegmatis* that were grown in brain heart infusion broth (BHI), or *C. albicans* cultured in Sabouraud liquid medium. All media were purchased from Oxoid.

### 2.2. Quinones and Others Antibacterial Compounds

Celastrol and pristimerin were isolated, purified, and characterized from the roots of Celastraceae species as previously reported [37,38]. Pure compounds were dissolved in dimethyl sulfoxide (DMSO) before the evaluation. The reference antibacterial agents ciprofloxacin, rifampicin, tetracycline, penicillin, and clofoctol (Sigma-Aldrich, St. Louis, MO, USA) were used as controls according to the Clinical and Laboratory Standards Institute [39].

### 2.3. Determination of Minimum Inhibitory Concentration (MIC) and Minimum Bactericidal Concentration (MBC)

The antimicrobial activity was determined for each compound in triplicate by broth microdilution method (range 0.08–40 μg/mL) in 96-well microtiter plates, according to the M07-A9 approved standard of the Clinical and Laboratory Standards Institute (CLSI) [40]. Wells with the same proportions of DMSO were used as controls and never exceeded 1% (*v*/*v*). The starting concentration of microorganism ranged between 1 and 5 × 10^5^ colony-forming units (CFU)/mL, and growth was monitored by measuring the increase in optical density at 550 nm (OD_550_) with a microplate reader (Infinite M200, Tecan Group Ltd., Männedorf, Switzerland) and viable count in agar plates. The minimum inhibitory concentration (MIC) was defined as the lowest concentration of compound that completely inhibits growth of the organisms compared to the untreated cells. All wells with no visible growth were subcultured in duplicate by transferring (100 µL) to nutrient, BHI, or Sabouraud agar plates. After overnight incubation, colony counts were performed and the minimum bactericidal concentration (MBC) was defined as the lowest compound concentration that produces ≥99.9% killing of the initial inoculum.

### 2.4. Bacterial Killing Assays

Overnight liquid cultures of *B. subtilis* were diluted in Monod’s flask containing 10 mL of NB medium, to give a working concentration ranging between 1 and 5 × 10^5^ CFU/mL. Celastrol (3 µg/mL) or pristimerin (10 µg/mL) was added at time 0 (lag phase) or after 3 h of incubation (log-phase, OD_550_ ≈ 0.4). These suspensions were incubated at 37 °C in a rotator shaker and growth was monitored by measuring the OD_550_ and CFU count on nutrient agar plates. Cultures with known antibiotics or without drugs were used as positive and negative controls, respectively. The assays were repeated at least three times.

### 2.5. Inoculum Effect

Overnight liquid cultures of *B. subtilis* were 10-fold serial diluted in NB medium to give different inoculum concentrations (10^3^ to 10^7^ CFU/mL). Celastrol (3 μg/mL) and pristimerin (10 μg/mL) were added and cultures were incubated at 37 °C in a rotatory shaker. Bacterial growth was monitored as described above. The assays were repeated at least three times.

### 2.6. Measurement of Radioactive Precursor Incorporation

Cultures of *B. subtilis* were diluted to obtain 10^6^ CFU/mL in Davis–Mingioli medium [41] supplemented with glucose (1%), asparagine (0.1 g/L), and casamino acids (2 g/L) (pH 7). The cultures were grown at 37 °C under shaking for at least 3 h to obtain an optical density (OD_550_) of 0.4. Volumes of 10 mL were transferred to prewarmed flasks containing celastrol (3 µg/mL) or pristimerin (10 μg/mL) and one of the precursors of DNA (1 μCi/mL [6-^3^H] + 2 μg/mL unlabeled thymidine), RNA (1 μCi/mL [5-^3^H] + 2 μg/mL unlabeled uridine), protein (5 μCi/mL [4,5-^3^H] + 2 μg/mL unlabeled leucine), or cell wall peptidoglycan (0.1 μCi/mL *N*-Acetyl-D-[1-^14^C] glucosamine). The samples were incubated at 37 °C under shaking. Volumes of 0.5 mL were collected and precipitated with 2 mL ice-cold 10% trichloroacetic acid (TCA) at different times. After 30 min of incubation in cold TCA, samples were filtered on glass microfiber filters grade GF/C (Whatman, Maidstone, UK)) and washed three times with 5 mL cold 10% TCA and once with 5 mL of 95% ethanol. The dried filters were placed in vials covered with a scintillation cocktail and counted in counter LKB Wallac Rackbeta (Perkin Elmer, Courtaboeuf, France). Macromolecular synthesis was measured by quantifying the incorporation of radiolabeled precursors (Amersham Biosciences Europe GmbH) into acid-insoluble material. Evaluations with DMSO added in the same proportion or a specific inhibitor of each biosynthetic pathway were included as negative and positive controls, respectively.

### 2.7. Measurement of Solutes Uptake

Solutes uptake was measured as total cell-associated counts after addition of quinones on cultures of *B. subtilis* growing exponentially (OD_550_ ≈ 0.2). Growing cells containing half concentration of NB medium were transferred to prewarmed flasks containing celastrol (3 µg/mL) or pristimerin (10 μg/mL) and radiolabeled glucose (2 μCi/mL D-[1-^14^C]-glucose) or the precursors of DNA, RNA, protein, and cell wall peptidoglycan (see concentrations in previous section). At different times (up to 30 min), 0.5 mL of samples was collected and filtered through Millipore filters of 0.45-μm pore size (Type HA, Millipore Corporation, Burlington, MA, USA). Filters were washed three times with 5 mL phosphate buffer, dried, and radioactivity was measured as above. Samples with clofoctol (5 µg/mL) or DMSO at the same concentration were included as positive and negative controls, respectively.

Furthermore, the effect of celastrol on the uptake and incorporation of radiolabeled precursors of DNA was also evaluated when the macromolecular synthesis was inhibited. *B. subtilis* cultures prepared as described above were treated for 30 min with the specific inhibitor ciprofloxacin (1.5 µg/mL) before the addition of labeled and unlabeled precursor. After 5 min, celastrol (3 µg/mL) or the same proportion of DMSO in control cultures was incorporated. At different times, precursor uptake and incorporation was measured as mentioned above. The assays were repeated at least three times.

### 2.8. Inhibition of DNA Gyrase

Inhibition of DNA gyrase was performed using the Gyrase Supercoiling kit 1 (#K001) (John Innes Enterprises Ltd., Norwich, UK) as described in the manufacturer’s instructions. Two units of DNA gyrase were incubated with 0.5 µg relaxed plasmid pBR322 and 50 µg/mL of celastrol or pristimerin in a final volume of 30 µL. Ciprofloxacin at 25 and 50 µg/mL was used as positive control. The samples were incubated at 30 °C for 30 min and the products were visualized in 0.8% agarose gel containing 0.5 µg/mL ethidium bromide.

### 2.9. Integrity of Cell Membrane

Bacterial membrane damage was examined by determination of the release of material absorbing at 260/280 nm, detection of potassium (K^+^) leakage, and the BacLight Live/Dead staining method (Invitrogen Molecular Probes, Eugene, OR, US). *B. subtilis* cultures in log-phase growth (OD_550_ ≈ 0.8) were centrifuged at 15,000× *g* for 10 min at 4 °C and washed twice with saline buffer. The pellet was resuspended in the same buffer to obtain a bacterial concentration of 1–2 × 10^8^ CFU/mL (or 1–2 × 10^7^ CFU/mL for K^+^ leakage experiments). Cell cultures were treated with celastrol (3 μg/mL) and incubated at 37 °C under shaking. Cultures with clofoctol (10 µg/mL) or DMSO at the same concentration were used as positive and negative controls, respectively. Samples were collected for quantification at different times over a 30-min period. Liberation of cytoplasmic materials was monitored by measuring the optical density (OD_260_ and OD_280_) of the supernatant after removing cells by centrifugation (at 9500× *g* for 10 min, 4 °C) or after membrane filtration by means of an atomic absorption spectrophotometer (Model Thermo S-Series, Thermo Electron Corporation, Cambridge, UK) for K^+^ release. The BacLight assay was analyzed after a 20-min dark-staining period following the manufacturer’s instructions. The cells were visualized at ×1000 magnification with an epifluorescence microscope (Leica DM4B, Leica Microsystems GmbH, Wetzlar, Germany) provided with a fluorescein–rhodamine dual filter.

### 2.10. Transmission Electron Microscopy

To further confirm the mode of action of celastrol on *B. subtilis*, transmission electron microscopy (TEM) analysis was performed. Suspensions of *B. subtilis* in log-phase growth (10^7^ CFU/mL) were treated with celastrol (3 µg/mL) for 1 h at 37 °C and were harvested at 6500× *g* for 8 min at 4 °C. For comparative purposes, bacterial cells grown under the same conditions were also treated with pristimerin (10 µg/mL). Subsequently, bacteria were washed in fixative buffer, post-fixed in 1% osmium tetroxide in fixative buffer, and washed with distilled water. Sections (1 µm) were cut with a Reichert Ultracut ultramicrotome and stained with toluidine blue; ultra-thin sections were contrasted with uranyl acetate and lead. Preparations were observed under a Zeiss EM 912 transmission electron microscope. Images were captured with a Proscan Slow-scan CCD-Camera for TEM (Proscan, Scheuring, Germany) and Soft Imaging System software (version 5.2, Olympus Soft Imaging Solutions GmbH, Münster, Germany). Control experiments with the same proportion of DMSO were performed in parallel.

### 2.11. Oxygen Consumption

Suspensions of *B. subtilis* and *E. coli* in the log phase of growth (OD_550_ ≈ 0.8) were centrifuged at 15,000× *g* for 10 min at 4 °C and washed twice with phosphate buffer 0.1 M (pH 7.0). Then, the pellet was suspended in the same buffer to obtain a bacterial concentration of 1–2 × 10^7^ CFU/mL. Cells suspensions (2.7 mL) supplemented with 0.3 mL of 10% glucose were used to measure oxygen consumption at room temperature in a glass cell of Clark oxygen electrode equipped with a magnetic stirrer. Celastrol (at 3 µg/mL for *B. subtilis* and 20 µg/mL for *E. coli*) was added to the cell suspension, and the steady-state output of the oxygen electrode (after 4 min) was measured using a digital biological oxygen monitor (model YSI 5300, Yellow Springs, OH, USA). DMSO in the same proportion and sodium cyanate at 6.7 mM were used as negative and positive controls, respectively. Furthermore, disrupted cell preparations of *B. subtilis* and *E. coli* were used to determine the effect of celastrol on oxygen uptake using NADH (0.1 mM) as a substrate. Cultures grown in yeast extract and peptone (YP, 1% *w*/*v*) medium for 18 h at 37 °C under aeration were collected by centrifugation at 10,000× *g* for 10 min at 4 °C. Cells were washed twice in 0.1 M potassium phosphate buffer (pH 7.0), resuspended in the same buffer (5 mL of buffer per gram of cells), and sonicated (Labsonic M, Sartorius Stedim Biotech, Göttingen, Germany) for 15 min in 10-s bursts with 20-s stop intervals. Intact cells were removed by centrifugation at 4000× *g* for 10 min at 4 °C. The supernatant was centrifuged at 15,000× *g* for 10 min at 4 °C and the pellet was resuspended in the same buffer as before (5 mL). Aliquots (0.2 mL) of disrupted cells were incubated at room temperature in the same buffer (2.8 mL) containing NADH (0.1 mM).

### 2.12. Statistical Analysis

Three independent experiments were conducted for each evaluation and means and standard deviations (±SD) were calculated. Analysis of variance (one-way ANOVA) followed by Tukey’s multiple comparison test (*p* < 0.05) to extract the specific differences between treatments were performed using R statistical software environment version 4.0.3 (R Foundation for Statistical Computing, Vienna, Austria).

## 3. Results and Discussion

### 3.1. Antimicrobial Activity

Table 1 shows the MICs and MBCs of celastrol and pristimerin against different microorganisms used in this study. Celastrol was active against all Gram-positive bacteria evaluated, with MIC values ranging between 0.16 (*B. subtilis*) and 2.5 µg/mL (*S. saprophyticus*). Compared to celastrol, pristimerin showed weaker activity on Gram-positive bacteria and no action on *S. aureus* and *M. smegmatis*. On the basis of these results, *B. subtilis* was selected to evaluate the mechanism of action of these quinones.

Gram-negative bacteria and the yeast *C. albicans* were insensitive to the action of both compounds at the maximum concentration assayed (40 µg/mL). The inactivity of pristimerin against *S. aureus* was previously reported by our group [24], although in later works, da Cruz et al. [42] indicated MIC values of 25 µg/mL. The use of a bacterial strain of *S. aureus* less sensitive to the activity of the compound could explain this difference in the results. Gullo et al. [23] reported antimicrobial activity for pristimerin against *C. albicans* with an MIC of 250 µg/mL. In our evaluations, the maximum concentration tested was 40 µg/mL since higher concentrations of both compounds led to solubility problems.

These products differ in the functional group in ring E (Figure 1). As previously reported for other methylene quinones and phenolic nor-triterpenes compounds, replacement of the carboxylic group present in celastrol on C-29 by a methyl ester group reduces the antibacterial activity [14,43]. Celastrol has shown interesting pharmacological activities, although inconveniences related to poor water solubility, high toxicity, or poor stability have also been described [29,44]. Structural modifications in the triterpene quinones scaffolds could be of interest to obtain derivatives with improved antimicrobial activities and to overcome the pharmacokinetic limitations of these compounds.

### 3.2. Effects of Methylene Quinones against B. subtilis

The time–kill curves of *B. subtilis* cultures showed a different behavior when celastrol and pristimerin were added at different growth stages (Figure 2). Addition of celastrol at 3 µg/mL in the lag phase (10^5^ CFU/mL) had a bactericidal effect similar to ciprofloxacin with ≥3 Log_10_ in CFU reduction after 9 h of incubation (Figure 2A). Rifampicin and tetracycline also had a bactericidal behavior in the first hours of treatment, although the cultures recovered after 24 h of incubation. A bacteriostatic action and regrowth was observed with penicillin. Compared to celastrol, pristimerin at 10 µg/mL had a bacteriostatic effect, producing a minor reduction in the CFU counts (<3 Log_10_). However, when pristimerin was incorporated in the log phase of growth (10^7^ CFU/mL), a bactericidal action was observed, reducing the bacterial population 4.2 Log_10_ in 2 h of treatment, with up to 99.9% dead after 24 h of incubation (Figure 2B). In addition, the drop in optical density values (OD_550_) revealed the bacteriolytic nature of this compound. Under these conditions of growth, celastrol had a bacteriostatic action, similar to that shown by ciprofloxacin, tetracycline, and rifampicin, with a clear regrowth of the culture after 24 h in the presence of rifampicin. Penicillin had a limited effect on the control of *B. subtilis* cells actively growing.

The action of the methylene quinones was also evaluated against different inoculum sizes of *B. subtilis* in lag-phase growth (Table 2). At the higher inoculum concentration (10^7^ CFU/mL), celastrol showed a bacteriostatic activity during the exposition time and a lower reduction in CFU/mL compared to the log phase of growth (Figure 2B). A bacteriostatic effect was also observed at 10^6^ and 10^5^ CFU/mL and bactericide at 10^4^ CFU/mL. Pristimerin showed a similar behavior to that obtained with celastrol. However, when this compound was added in the log phase of growth, a bacteriolytic effect was observed within 6 h of treatment (Figure 2B). These results indicate that both quinones show a stronger effect when *B. subtilis* is actively growing.

### 3.3. Mechanism of Action

#### 3.3.1. Effects of Macromolecular Synthesis and Initial Uptake of Solutes

Initially, the incorporation of radiolabeled precursors into DNA, RNA, protein, and cell wall synthesis was measured. After addition of celastrol and pristimerin, the incorporation of all precursors into the macromolecular synthesis was blocked, but not simultaneously (Figure 3). Celastrol at 3 µg/mL reduced by ≥70% the incorporation of [6-^3^H] thymidine and [5-^3^H] uridine within 5 and 10 min, respectively. The inhibitory effect of celastrol after 5 min was comparable to that observed with the specific inhibitor of the RNA synthesis, rifampicin. Pristimerin at 10 µg/mL needed at least 30 min to produce an inhibition of 57% and 47% of DNA and RNA synthesis, respectively (Figure 3A,B). Celastrol and pristimerin inhibited the incorporation of leucine into protein synthesis by >55% after 20 min, whereas tetracycline blocked this process (>70%) in 10 min. The incorporation of *N*-acetyl-d-[^14^C] glucosamine into peptidoglycan decreased rapidly within 2 min after the addition of celastrol and pristimerin, with inhibition values of 58% and 70%, respectively. However, this effect was not constant over time and the incorporation of the precursor gradually increased after 5 min. Initially, the inhibitory effect produced by penicillin at 30 µg/mL on cell wall synthesis was slower (32% in 2 min) than with celastrol and pristimerin, but the incorporation of the precursor gradually increased this up to 85% in 30 min (Figure 3D). Clofoctol, a cytoplasmic membrane disruptor [45], had a variable effect on the incorporation of precursors but blocked all biosynthetic processes (>70% of inhibition) after 30 min of evaluation.

The inhibition of all processes of macromolecular synthesis is more compatible with an indirect effect on biosynthetic pathways rather than a specific action on specific targets [46,47]. Thus, the inhibition produced by celastrol and pristimerin on macromolecular synthesis could be related with damage on the cytoplasmic membrane, as it happens with clofoctol. For this reason, we firstly determined the effect of both compounds on the uptake of glucose by *B. subtilis*, measured as total cell-associated counts after cell isolation from free labeled precursors in the incubation medium. As shown in Figure 4, celastrol at 3 µg/mL drastically reduced (>70%) the uptake of D-[1-^14^C]-glucose in only 2 min, whereas clofoctol needed at least 5 min to produce comparable reductions. Pristimerin weakly reduced the uptake of glucose and required up to 20 min to produce ≈50% inhibition.

Based on these results, we decided to investigate whether the uptake of radiolabeled precursors would also be blocked in the presence of terpenoids, as was the case with glucose. The uptake of precursors (thymidine, uridine, leucine, and N-acetyl glucosamine) was determined in the presence of celastrol, the triterpenoid that showed the greatest inhibitory effect on glucose uptake and macromolecular synthesis. Figure 5 shows how the addition of celastrol at 3 µg/mL to *B. subtilis* cultures rapidly inhibited the uptake of [6-^3^H] thymidine and [5-^3^H] uridine by >50% between 2 and 5 min. The specific inhibitors of DNA and RNA synthesis, ciprofloxacin and rifampicin, did not affect the uptake of these precursors. The uptake of [4,5-^3^H] leucine was inhibited by up to 43% after 20 min, while uptake of *N*-acetyl-d-[1-^14^C] glucosamine was weakly affected and gradually increased during the experimentation time (Figure 5C,D). The addition of tetracycline blocked 75% the uptake of leucine in 10 min, whereas the transport of *N*-acetyl-d-[1-^14^C] glucosamine into the cells was not affected in the presence of penicillin.

Our results indicate that both uptake and incorporation processes in *B. subtilis* were affected by celastrol. Inhibition of macromolecular synthesis implies a low demand for biosynthetic precursors. Under these conditions, it would be reasonable to expect a slowdown in the uptake of precursors into the cell if both processes are coupled. In contrast, a primary blockage in precursor transport into the cells would also lead to inhibition of biosynthetic pathways due to low precursor availability. However, the uptake of precursors does not necessarily have to be linked to their incorporation in macromolecular synthesis. As Chou and Pogell [48] have previously suggested, if transport is inhibited and the process is not tightly coupled to macromolecular synthesis, the initial accumulation of precursors into the acid-soluble pool should be much more inhibited than their incorporation during macromolecular synthesis. In fact, these authors described how panamycin, an antibiotic that affects membrane-associated cellular functions, inhibits the uptake of uridine, while incorporation into biosynthetic processes is minimally affected. Thus, an ideal scenario would be one in which all of the precursors taken up by cells are in acid-soluble form to determine to what extent a blockage in molecular transport is a key process in the mechanism of action of the antibacterial agent [48].

Therefore, an experiment was conducted to investigate the effect of celastrol on the uptake of radiolabeled thymidine in *B. subtilis* when DNA synthesis was blocked by ciprofloxacin (1.25 µg/mL during 30 min). As seen in Figure 6, total radioactivity in control cells increased gradually throughout the incubation time. As expected, the incorporation of precursor slowed down when the biosynthetic process was specifically inhibited by ciprofloxacin. In this case, 5 min after the addition of the radiolabeled precursor, only ≈29% of available thymidine was incorporated into macromolecular synthesis. By contrast, in the absence of ciprofloxacin, ≈74% of the precursor present in the cells was incorporated in the same time (Figure 5A). This observation suggests that uptake and incorporation are independent processes and that the precursors can accumulate in the cytoplasm in the absence of incorporation. The addition of celastrol at 3 µg/mL not only blocked the uptake of thymidine radiolabeled precursor in the treated cells but also produced a slight leakage of accumulated [6-^3^H] thymidine from *B. subtilis*. This effect could be related to an action of celastrol on the integrity of the membrane, affecting its permeability and producing a release of the cytoplasmic content. Similar results have been observed on *S. aureus* cells treated with zeylasterone, another triterpenoid with antibacterial properties, for which an action on the cytoplasmic membrane has also been suggested [49]. Additionally, the arrest of uptake was accompanied by an immediate blockage of incorporation into macromolecular synthesis. The results obtained reinforce the idea that celastrol could first target the function of the cytoplasmic membrane by affecting the transport of solutes and other essential molecules into the cell.

#### 3.3.2. Effect of Celastrol on the Integrity and Functions of the Cytoplasmic Membrane

The cytoplasmic membrane is a delicate and metabolically active structure essential for the survival of microorganisms. It acts as a selective permeability barrier and prevents the loss of essential components of low molecular weight and nucleotide. Antibacterial agents targeting the cytoplasmic membrane affect its functions and cause a rapid release of low molecular weight compounds [50]. It has been described that the primary target site of the phenolic compounds in bacteria is the cytoplasmic membrane [51]. Damage to the cytoplasmic membrane impacts permeability barrier functions, which subsequently leads to a loss of structural integrity and a leakage of intracellular material. Thus, we also investigated the effect of celastrol on the cytoplasmic membrane of *B. subtilis* using (i) the BacLight test, (ii) detection of UV-absorbing material efflux, and (iii) determination of potassium leakage. Fluorescent dye and “LIVE/DEAD” BacLight Bacterial Viability Kits have the capability of monitoring the viability of bacteria as a function of the cell membrane integrity [52,53]. Surprisingly, microscopic observations after the BacLight assay showed that the cells treated with celastrol maintained membrane integrity like the untreated cells. In contrast, cultures treated with clofoctol showed red fluorescence, an observation indicating membrane damage (Appendix A
Appendix A). When UV-absorbing material from *B. subtilis* cultures was monitored, concentrations up to 20 µg/mL of celastrol did not alter the UV spectrum in comparison to the untreated cultures (Appendix A
Appendix A). Clofoctol used as a control clearly induced the release of UV-absorbing nucleotides in treated cells. Similar results were previously observed with netzahualcoyone, a terpenoid that interacts with the cytoplasmic membrane but also does not release materials that absorb at 260/280 nm [21]. We also determined the potassium released by *B. subtilis* cells as the first index of membrane damage [54]. Exposure to celastrol for 5 min induced intracellular potassium release compared with untreated cells, although the effect was significantly weaker than that observed with clofoctol (Figure 7). These data suggest that celastrol could act on biological membranes, altering their functions and modifying cell permeability.

#### 3.3.3. Transmission Electron Microscopy

To determine whether celastrol induces noticeable cell membrane damage, transmission electron microscopy was performed on thin sections of *B. subtilis* treated with the terpenoid for 1 h. Compared to the untreated control, cells treated with celastrol exhibited abnormally long cells, variability in wall thickness, compact ribosomes underlying the plasma membrane, and mesosome-like structures arising from the septa (Figure 8).

Despite the multiple effects produced by celastrol, the technique did not allow the observation of visible damage to the cell membrane. For comparison purposes, *B. subtilis* treated with pristimerin at 10 µg/mL, which showed a bacteriolytic effect during killing curves assays, was also observed. As with celastrol, the cells were extremely long, spindle-shaped, with thin cell walls and slightly electrodense cytoplasm with ribosomes associated to the inner face of the membrane (Figure 8D,E). These observations resemble those obtained by da Cruz et al. [42] on cultures of *S. aureus* treated with pristimerin, where the cells presented a disrupted membrane, as well as a loss of cell integrity.

#### 3.3.4. Effect of Celastrol on Cellular Respiration

Another fundamental function of the cytoplasmic membrane is cellular respiration. The respiratory chain plays an important role in the energetic metabolism of a cell, the maintenance of intracellular redox balance, or the protection against oxidative stress [55]. The membranes harbor the electron transport chain, a well-known system with oxygen as the final electron acceptor in aerobic respiration processes.

Thus, the effect of celastrol on oxygen consumption was also evaluated on cell cultures of *B. subtilis* and *E. coli*, as well as in acellular preparations of these bacteria obtained by cell disruption. Table 3 summarizes the results of the effect of celastrol on glucose-dependent oxygen uptake in intact cells of *B. subtilis* and *E. coli*. Both celastrol and NaCN produced an immediate inhibition of oxygen consumption in *B. subtilis*, reaching 60% at 8 min after their addition compared to the untreated control. As expected, celastrol barely affected the oxygen uptake in *E. coli*, as Gram-negative bacteria are insensitive to the terpene quinone.

The same evaluation was carried out on acellular preparations where the oxygen consumption was coupled to the oxidation of NADH. Unlike when whole cells were used, here, NADH oxidation only depended on the amount of dissolved oxygen and the function of the respiratory chain. In these conditions, the addition of celastrol and NaCN inhibited the oxygen consumption in *B. subtilis* and *E. coli* preparations. The different behavior shown by celastrol in intact and disrupted preparations of *E. coli* cells shows that the cell membranes of both Gram-positive and -negative bacteria are equally sensitive to its action. The outer membranes of Gram-negative bacteria act as a permeability barrier, holding celastrol physically distant from its target of action. These results indicate that celastrol has a direct effect on the electron transport chain, affecting the consumption of oxygen and, consequently, the oxidation of NADH associated with respiration processes. However, in the presence of celastrol, neither a spontaneous oxidation of NADH nor a reduction in NAD^+^ was observed.

An additional experiment was carried out to verify previous results indicating an action of celastrol on the inhibition of enzymatic activity. Nagase et al. [56] reported that celastrol can inhibit topoisomerase II and trigger apoptosis in HL-60 cells. DNA gyrase is an important bacterial topoisomerase II that catalyzes ATP-dependent negative supercoiling of bacterial DNA. The essential role of gyrase is to maintain the topological constitution of DNA and, hence, the survival of bacteria [57]. Eukaryotic cells lack the enzyme, which has led to the development of specific antimicrobials targeting the gyrase functions [58]. Our results confirmed the effect of celastrol at 50 µg/mL on the gyrase activity of supercoiling plasmid pBR322, as did ciprofloxacin, used as a positive control (Appendix A
Appendix A). Interestingly, pristimerin has shown a weaker antibacterial action compared to celastrol and did not affect the enzymatic activity of gyrase. This mechanism of action has been suggested for other triterpenoid compounds showing activity against topoisomerases I and II [59]. Some models have been proposed for the mode of action of triterpenoids on topoisomerases, which can be either by binding to the enzyme at the DNA binding site or by binding to the ATP binding site, conformationally blocking DNA binding to the enzyme [60]. Thus, the effect on DNA supercoiling produced by celastrol could be related with an action on the gyrase rather than a direct DNA binding effect.

## 4. Conclusions

The antimicrobial action of celastrol and pristimerin, two natural triterpene methylene quinones, was evaluated, and the mechanism of action against the spore-forming bacteria *Bacillus subtilis* was also approached. Celastrol showed a higher antimicrobial effect compared with pristimerin, being active against Gram-positive bacteria. The results obtained in this study indicate that celastrol interacts with the cytoplasmic membrane of *B. subtilis*, preventing the transport of solutes into the cells and affecting basic membrane functions such as respiration processes. At the structural level, the membrane was not widely affected, suggesting a mechanism of action for celastrol other than a simple effect on structural components and subsequent membrane disruption. Furthermore, celastrol can also interact with enzymatic processes different from those exclusively located on the cytoplasmic membrane, as observed here for topoisomerase II. Although further investigations are required to elucidate a more precise mechanism of action, our results indicate that celastrol can act on multiple targets.

## Figures and Tables

**Figure 1 foods-10-00591-f001:**
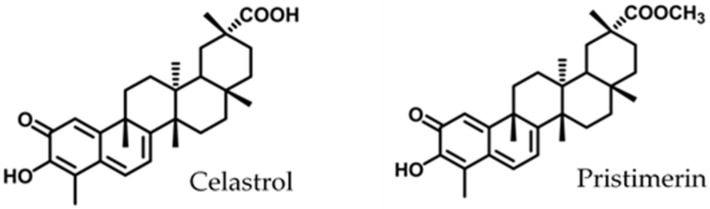
Structure of triterpenoid methylene quinones.

**Figure 2 foods-10-00591-f002:**
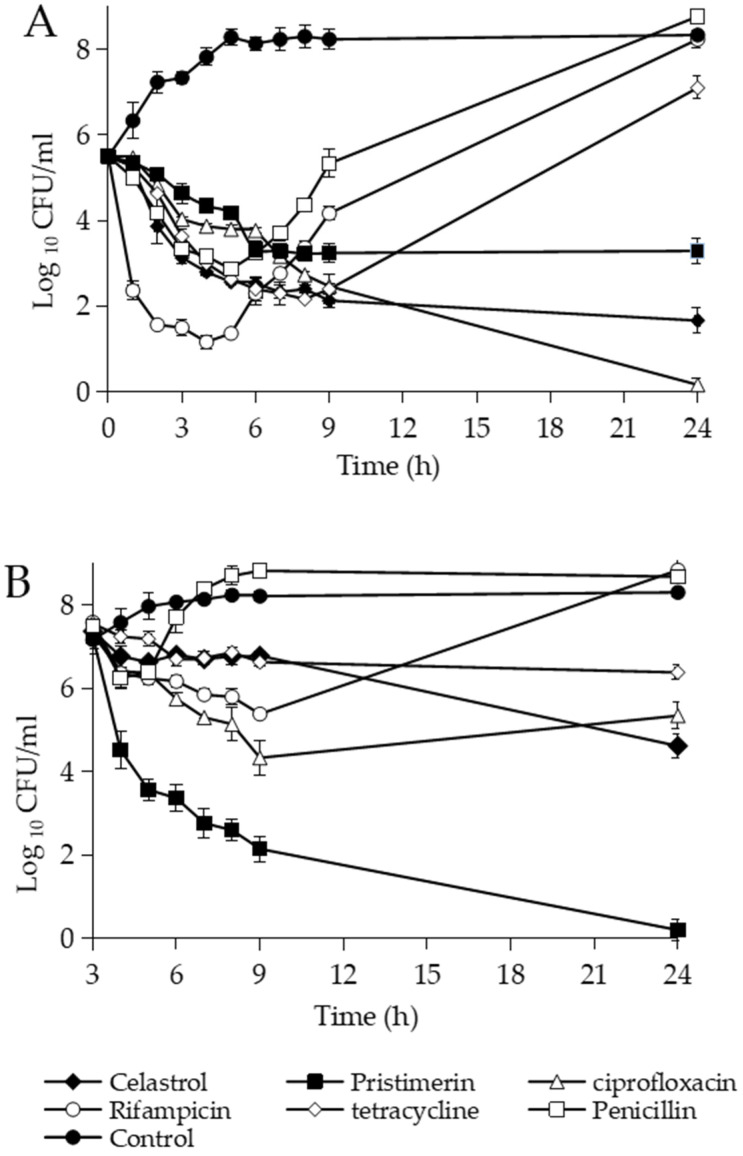
Time–kill curves of *B. subtilis* cultures expressed as Log_10_ of CFU counts after treatment with different antimicrobial substances (celastrol 3 µg/mL; pristimerin 10 µg/mL; ciprofloxacin 1.2 µg/mL; rifampicin 0.15 µg/mL; tetracycline 7.5 µg/mL; and penicillin 9 µg/mL) added in lag phase of growth (**A**) and log phase after three hours of preincubation (**B**). Cultures without drugs and with the maximum proportion of DMSO were used as controls. Error bars express SD with *n* = 3.

**Figure 3 foods-10-00591-f003:**
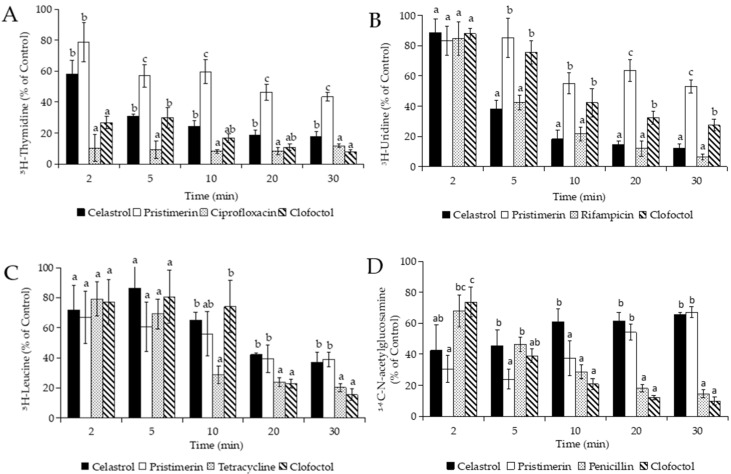
Incorporation of precursor in the synthesis of DNA (**A**), RNA (**B**), protein (**C**), and cell wall (**D**) of *B. subtilis* cultures in the presence of triterpene methylene quinones (celastrol 3 μg/mL; pristimerin 10 μg/mL), specific inhibitors of each pathway (ciprofloxacin 1.25 μg/mL; rifampicin 0.2 µg/mL; tetracycline 10 µg/mL; and penicillin 30 µg/mL), and clofoctol (5 µg/mL). Data are expressed as percentage (%) of precursors’ incorporation compared to controls without drugs but with the maximum proportion of DMSO. Error bars express SD with *n* = 3. Different letters above bars mean significant differences between treated cultures (*p* < 0.05, one-way ANOVA; Tukey’s test).

**Figure 4 foods-10-00591-f004:**
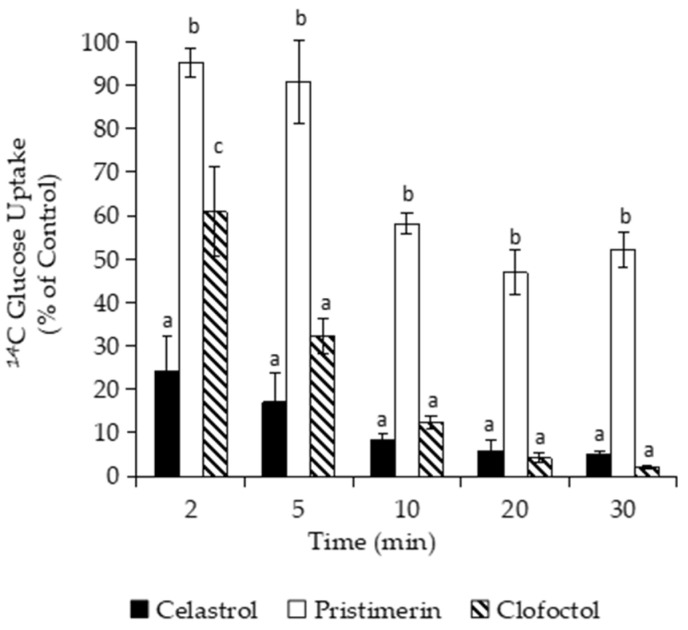
Glucose uptake assay on *B. subtilis* cultures after the addition of the quinones celastrol (3 µg/mL) and pristimerin (10 µg/mL). Clofoctol (5 µg/mL), a known inhibitor of macromolecule uptake, was added as a positive control. Data are expressed as percentage (%) of precursor incorporation compared to controls without drugs but with the maximum proportion of DMSO. Error bars express SD with *n* = 3. Different letters above bars mean significant differences between treated cultures (*p* < 0.05, one-way ANOVA; Tukey’s test).

**Figure 5 foods-10-00591-f005:**
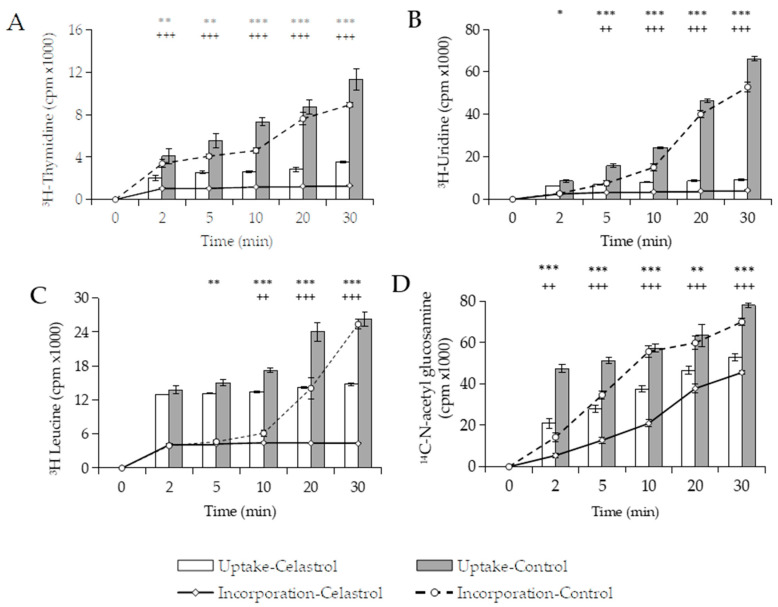
Incorporation (insoluble phase) and uptake (soluble phase) of DNA (**A**), RNA (**B**), protein (**C**), and cell wall (**D**) precursors on *B. subtilis* cultures in the presence of celastrol (3 μg/mL). Cultures in the same conditions but without celastrol and with the same proportion of DMSO were used as negative control. Error bars express SD with *n* = 3. Significant differences found in uptake (***: *p* < 0.001. **: *p* < 0.01. *: *p* < 0.05) or incorporation (+++: *p* < 0.001. ++: *p* < 0.01) compared with control (one-way ANOVA; Tukey’s test).

**Figure 6 foods-10-00591-f006:**
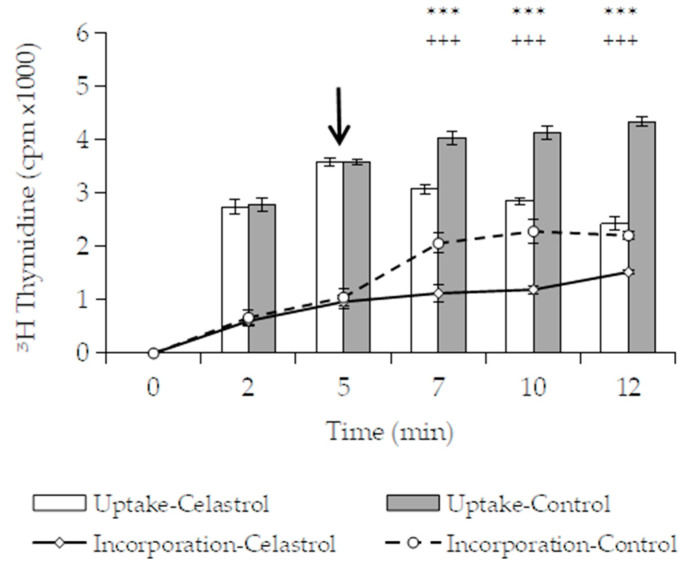
Incorporation (insoluble phase) and uptake (soluble phase) of radiolabeled ^3^H-thymidine on *B. subtilis*. Bacterial culture were pretreated with ciprofloxacin (1.25 μg/mL), a specific inhibitors of DNA biosynthesis before the addition of the radiolabeled precursor (time 0). The triterpene celastrol (3 μg/mL), or the same proportion of DMSO used as control, was added at the time indicated by the arrow. Error bars express SD with *n* = 3. Significant differences found in uptake (***: *p* < 0.001) or incorporation (+++: *p* < 0.001) compared with control (one-way ANOVA; Tukey’s test).

**Figure 7 foods-10-00591-f007:**
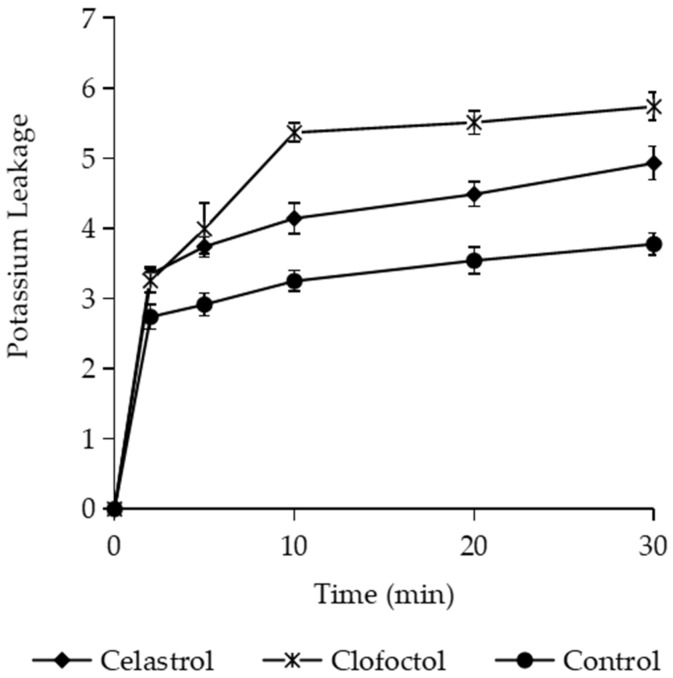
Potassium (K^+^) leakage (μg/mL) from *Bacillus subtilis* cells exposed to celastrol (3 μg/mL). Cultures with clofoctol (5 µg/mL) or the maximum proportion of DMSO were used as positive and negative controls, respectively. Error bars express SD with *n* = 3.

**Figure 8 foods-10-00591-f008:**
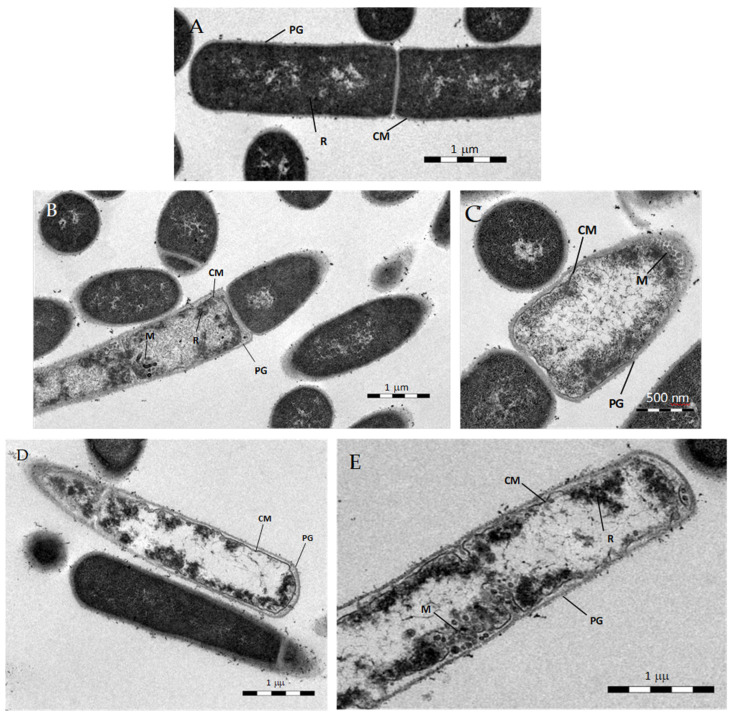
TEM of *B. subtilis* exposed for 1 h to celastrol at 3 μg/mL (**B**,**C**) or pristimerin at 10 μg/mL (**D**,**E**). Untreated cells (**A**). CM, cytoplasmic membrane; PG, peptidoglycan layer; R, ribosomes; M, mesosome-like structures.

**Table 1 foods-10-00591-t001:** Minimum inhibitory concentration (MIC) ^1^ and minimum bactericidal concentration (MBC) ^1^ of celastrol and pristimerin, expressed in μg/mL against different microorganisms ^2^.

Microorganisms	Celastrol		Pristimerin	
	MIC	MBC	MIC	MBC
*Bacillus subtilis*	0.156	2.5	1.25	10
*B. cereus*	0.625	2.5	10	>40
*B. pumilus*	0.625	2.5	2	10
*B. megaterium*	1.25	5	20	>40
*Staphylococcus aureus*	1.25	>40	>40	>40
*S. epidermidis*	0.312	15	1.25	>40
*S. saprophyticus*	2.5	10	10	10
*Mycobacterium smegmatis*	5	>40	>40	>40
*Enterococcus faecalis*	1.25	40	20	>40

^1^ Values represent average obtained from a minimum of three experiments. ^2^ The quinone compounds were inactive against Gram-negative bacteria and the yeast assayed (MIC > 40 µg/mL).

**Table 2 foods-10-00591-t002:** Effect of celastrol at 3 μg/mL and pristimerin at 10 μg/mL on different inoculum sizes of *B. subtilis* at 3 and 6 h after treatment.

Triterpene	Dilution Factor	Mean Log_10_ CFU ± SD ^1^		
		Initial Inoculum	Recovery at 3 h	Recovery at 6 h
Celastrol	10^7^	7.58 ± 0.11	7.60 ± 0.15	7.58 ± 0.05
	10^6^	6.41 ± 0.14	4.68 ± 0.27	4.08 ± 0.17
	10^5^	5.54 ± 0.20	4.00 ± 0.21	3.15 ± 0.16
	10^4^	4.67 ± 0.09	2.30 ± 0.25	1.27 ± 0.25
Pristimerin	10^7^	7.58 ± 0.12	7.51 ± 0.07	7.47 ± 0.05
	10^6^	6.35 ± 0.17	5.32 ± 0.20	4.71 ± 0.33
	10^5^	5.32 ± 0.19	4.26 ± 0.16	3.27 ± 0.22
	10^4^	4.75 ± 0.11	3.59 ± 0.12	1.56 ± 0.10

^1^ Data are expressed as mean values ± standard deviations (*n* = 3).

**Table 3 foods-10-00591-t003:** Mean values ± standard deviations of means (*n* = 3) of oxygen consumption rates (µL O_2_/min) at different times in whole cells or membrane preparations of *B. subtilis and E. coli* without treatment or after treatments with celastrol at 3 µg/mL or sodium cyanide (NaCN) at 6.7 mM used as positive controls. The percentage (%) of inhibition in the oxygen consumption referring to the untreated cells (negative control) is shown in parentheses.

Bacteria	Time (min)	Oxygen Compsumption ^1^					
		Whole Cells (Glucose 1%)			Membrane Fraction (NADH 0.1 mM)		
		Control	NaCN	Celastrol	Control	NaCN	Celastrol
*B. subtilis*	0	2.36 ± 0.10 ^a^	2.30 ± 0.07 ^a^ (2.4%)	2.27 ± 0.12 ^a^ (3.9%)	0.54 ± 0.05 ^a^	0.50 ± 0.06 ^a^ (6.9%)	0.52 ± 0.08 ^a^ (4.0%)
	2	3.69 ± 0.17 ^a^	2.71 ± 0.05 ^b^ (26.5%)	2.74 ± 0.15 ^b^ (25.7%)	0.98 ± 0.11 ^a^	0.66 ± 0.00 ^b^ (31.7%)	0.68 ± 0.01 ^b^ (30.6%)
	4	5.42 ± 0.46 ^a^	2.77 ± 0.08 ^b^ (48.6%)	3.11 ± 0.04 ^b^ (42.4%)	1.27 ± 0.01 ^a^	0.79 ± 0.00 ^c^ (37.7%)	0.82 ± 0.02 ^b^ (35.1%)
	6	7.09 ± 0.21 ^a^	2.91 ± 0.03 ^c^ (58.9%)	3.30 ± 0.02 ^b^ (53.5%)	1.56 ± 0.05 ^a^	0.89 ± 0.01 ^b^ (43.0%)	0.94 ± 0.03 ^b^ (39.4%)
	8	8.53 ± 0.14 ^a^	2.97 ± 0.02 ^c^ (65.2%)	3.42 ± 0.04 ^b^ (59.8%)	1.80 ± 0.01 ^a^	0.94 ± 0.00 ^c^ (47.9%)	1.01 ± 0.02 ^b^ (43.9%)
*E. coli*	0	2.16 ± 0.07 ^a^	2.04 ± 0.06 ^a,b^ (5.5%)	2.00 ± 0.02 ^b^ (7.3%)	2.22 ± 0.07 ^a^	2.10 ± 0.01 ^a,b^ (5.0%)	2.07 ± 0.05 ^b^ (6.5%)
	2	3.35 ± 0.14 ^a^	2.16 ± 0.03 ^c^ (35.5%)	3.08 ± 0.08 ^b^ (7.9%)	3.07 ± 0.00 ^a^	2.20 ± 0.05 ^c^ (28.2%)	2.41 ± 0.02 ^b^ (21.3%)
	4	4.34 ± 0.24 ^a^	2.22 ± 0.03 ^b^ (48.8%)	3.95 ± 0.14 ^a^ (9.0%)	3.88 ± 0.01 ^a^	2.30 ± 0.03 ^c^ (40.6%)	2.51 ± 0.01 ^b^ (35.3%)
	6	5.80 ± 0.32 ^a^	2.32 ± 0.03 ^c^ (60.0%)	4.84 ± 0.10 ^b^ (16.6%)	4.74 ± 0.01 ^a^	2.34 ± 0.02 ^c^ (50.6%)	2.61 ± 0.02 ^b^ (44.9%)
	8	6.79 ± 0.19 ^a^	2.40 ± 0.05 ^c^ (64.6.4%)	5.53 ± 0.09 ^b^ (18.5%)	5.37 ± 0.00 ^a^	2.40 ± 0.08 ^c^ (55.3%)	2.68 ± 0.01 ^b^ (50.0%)

^1^ For each assessment, values with different superscript letters within each given time point indicate statistically significant differences (ANOVA, Tukey’s multiple comparison test, *p* < 0.05).

## Data Availability

Not applicable.

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
