# Peer review of "Antimicrobial Activity and Mode of Action of Celastrol, a Nortriterpen Quinone Isolated from Natural Sources"

_foods, 2021, doi:10.3390/foods10030591_

Round 1

Reviewer 1 Report

1. Introduction section:

-Please include the rational for choosing such bacterial strains for the tests.

-Did the author use a bioinformatics tool?

2. Table 1 : please include S.D. values.

3. Figures 3-6: Please include the ANOVA P values in the legends.

4. The indication of statistically significant values are misleading. Different formatting is used in figures and tables. Please use a unique uniform.

Author Response

Dear reviewer,
I am very grateful for our valuable comments. We tried to address all required
questions (below). We hope the answers can meet your expectations.
Corresponding authors
1. Introduction section:
- Please include the rational for choosing such bacterial strains.
Answer: B. subtilis is representative species of the Bacillus genus. In addition, it was chosen
because it was the bacterium that showed the highest sensitivity to the evaluated
nortriterpene quinones. We have included in the results and discussion section a sentence to
clarify why this bacterium was selected.
- Did the author use a bioinformatics tool?
Answer: We have not used bioinformatic tools for the research presented in this manuscript.
2. Table 1: please include S.D. values
Answer: The MICs have been determined according the CLSI using concentrations derived
traditionally from two fold dilutions indexed to the base 1 (e.g. 1,2,4, 8, 16 µg/mL,…etc).The
MIC is the lowest concentration of antimicrobial agent that completely inhibits growth of the
organism in the tubes or microdilution wells as detected by unaided eye or measured by
optical density (mainly microdilution method). Therefore, standard deviation (SD) is not
applicable here since, the MIC values cannot be accurately determined and should be reporter
as equal two or less than the lowest concentration tested. It would be applicable for MIC50
(for antifungal activity) since it is referred to the minimum concentration that inhibits 50% of
the growth respect to control. In this particular case the results are determined from the
inhibition percentage.
3. Figures 3-6: Please include the ANOVA P values in the legends.
Answer: We have indicated in the figure legends the level of significance (p value) of each
analysis using the standard cut-off point α = 0.05 for Figures 3 and 4 (significant differences
between treatments) and up to α = 0.01 and α = 0.001 for figures 5 and 6 (significant
differences of one treatment compared to the control).
To avoid unnecessary overload in the legend of Figure 6, cut off α values of 0.05 and 0.01 were
omitted because the entire analysis allowed differences to be established with α = 0.001.
4. The indication of statistically significant values are misleading. Different formatting is
used in figures and tables. Please use a unique uniform.
Answer: Statistical analysis in Figures 3 and 4 shows differences between treatments. These
graphs do not represent untreated (control) cultures in which 100% incorporation is 
considered. The data in the graph are shown as a percentage of inhibition in the incorporation
of precursors as a consequence of the treatments (evaluated compounds) referred to the
control (0% inhibition). The objective of this analysis was to establish the different behaviours
between treatments; that is, how different (or similar) the compounds under evaluation
(triterpenes) are from each other or compared to a specific known inhibitor of each pathway.
In this case, the differences are identified by different letters with α = 0.05 as the cut-off point.
In contrast, in Figure 5 and Figure 6, only one triterpene (celastrol) is compared to the
untreated control (no drug). The result is a combined graph ("dual" representation) where
two different parameters (incorporation and uptake) are evaluated. Statistical analyses
represent the differences of treated and untreated cultures for both parameters (represented
as * and +, respectively). During the preparation of the manuscript we tried to represent
significant differences with letters in order to unify criteria with the other figures of the work.
However, the results were unclear and difficult to interpret. At the same time, the symbols
allowed us to include a more restrictive analysis by adding an α cut off of 0.01 and 0.001 in
addition to the 0.05 standard that is generally used in most studies.
Although we would like to include all the reviewers' suggestions, we consider that unifying
the criteria for the entire statistical analysis (same format in all graphs and tables) could
generate some problems in clearly understanding the data shown in Figures 5 and 6.
Therefore, we would like to ask the reviewer for the possibility of keeping the current format.
If, even so, it is still considered that there is a need to change the format in which the data is
represented, we would proceed to modify it. 

Reviewer 2 Report

Specific comments may be found in the attached document.

General comments: The authors have produced an interesting paper, which has covered a lot of material.  The manuscript is well-organized and the information presented as in figures and tables hence making it easy to follow.  Further, addressing the detailed comments in the document will greatly improve the manuscript overall.

Author Response

Dear reviewer,
We are very grateful for the comments.
Following the reviewer’s recommendations, we have corrected the grammar, style and
some mistakes made in the first version.
We also tried to answer questions asked by the reviewer when necessary. See below
for a detailed list of changes/modifications we made.
Corresponding author
Question: Page 2, line 5 Why was this species chosen over the others?
Answer: B. subtilis was chosen because it was the bacterium that showed the highest
sensitivity to the evaluated nortriterpen quinones.
Some acronyms were defined throughout the manuscript
- Page 2, line 93. The meaning of CLSI (Clinical and Laboratory Standards
Institute) was added.
- Page 4 line 190. We added osmium tetroxide when necessary.
- Page 5 line 279. We do not consider it necessary to add the meaning of
NADH.
- Page 5 line 280. We added the meaning YP medium - Yeast extract and
peptone (1% w/v)
Grammar, style, mistakes have been corrected.
Page 3, line 131. Counter instead of counted
Page 5, Table 2. 2.5 instead of 2,5
Page 6, line 318. Shown instead of showed
Page 6, line 319. To instead of with
Page 6, line 338. That shown instead of the showed
Page 7, line 357. That instead of the
Page 8, line 370. “The” was added
Page 8, line 370. Were instead of was
Page 8, line 373. That instead of the
Page 8, line 377. Blocked instead of blockage
Page 9, line 405. Needed instead of need
Page 10, line 422. DNA instead of AND
Page 10, line 426. The sentence was rewritten
Page 10, line 443. The sentence was rewritten
Page 12, line 484. Additional reference was not added because whole paragraph is
covered by the reference [50]
Page 12, line 497. Indicating instead of compatible with
Page 16, line 574. Which instead of what
Page 16, line 578. The sentence was rewritten
Page 16, line 584. The sentence was rewritten

This manuscript is a resubmission of an earlier submission. The following is a list of the peer review reports and author responses from that submission.